# Beneficial Effect of Mildly Pasteurized Whey Protein on Intestinal Integrity and Innate Defense in Preterm and Near-Term Piglets

**DOI:** 10.3390/nu12041125

**Published:** 2020-04-17

**Authors:** Marit Navis, Vanesa Muncan, Per Torp Sangild, Line Møller Willumsen, Pim J. Koelink, Manon E. Wildenberg, Evan Abrahamse, Thomas Thymann, Ruurd M. van Elburg, Ingrid B. Renes

**Affiliations:** 1Tytgat Institute for Intestinal and Liver Research, Amsterdam Gastroenterology Endocrinology and Metabolism, Amsterdam UMC, Location AMC, University of Amsterdam, 1105 AZ Amsterdam, The Netherlands; m.navis@amsterdamumc.nl (M.N.); v.muncan@amsterdamumc.nl (V.M.); p.j.koelink@amsterdamumc.nl (P.J.K.); m.e.wildenberg@amsterdamumc.nl (M.E.W.); 2Comparative Pediatrics & Nutrition, University of Copenhagen, DK-1870 Copenhagen, Denmark; pts@sund.ku.dk (P.T.S.); willumsen.lm@gmail.com (L.M.W.); thomas.thymann@sund.ku.dk (T.T.); 3Department of Neonatology, Rigshospitalet, DK-2100 Copenhagen, Denmark; 4Department of Pediatrics, Odense University Hospital, DK-5000 Odense, Denmark; 5Danone Nutricia Research, 3585 CT Utrecht, The Netherlands; Evan.Abrahamse@danone.com; 6Laboratory of Food Chemistry, Wageningen University, 6708 PD Wageningen, The Netherlands; 7Emma Children’s Hospital, Amsterdam UMC Location AMC, 1105 AZ Amsterdam, The Netherlands; rm.vanelburg@amsterdamumc.nl

**Keywords:** intestinal maturation, intestinal barrier, innate defense, preterm piglets, whey protein

## Abstract

Background. The human digestive tract is structurally mature at birth, yet maturation of gut functions such as digestion and mucosal barrier continues for the next 1–2 years. Human milk and infant milk formulas (IMF) seem to impact maturation of these gut functions differently, which is at least partially related to high temperature processing of IMF causing loss of bioactive proteins and formation of advanced glycation end products (AGEs). Both loss of protein bioactivity and formation of AGEs depend on heating temperature and time. The aim of this study was to investigate the impact of mildly pasteurized whey protein concentrate (MP-WPC) compared to extensively heated WPC (EH-WPC) on gut maturation in a piglet model hypersensitive to enteral nutrition. Methods. WPC was obtained by cold filtration and mildly pasteurized (73 °C, 30 s) or extensively heat treated (73 °C, 30 s + 80 °C, 6 min). Preterm (~90% gestation) and near-term piglets (~96% gestation) received enteral nutrition based on MP-WPC or EH-WPC for five days. Macroscopic and histologic lesions in the gastro-intestinal tract were evaluated and intestinal responses were further assessed by RT-qPCR, immunohistochemistry and enzyme activity analysis. Results. A diet based on MP-WPC limited epithelial intestinal damage and improved colonic integrity compared to EH-WPC. MP-WPC dampened colonic IL1-β, IL-8 and TNF-α expression and lowered T-cell influx in both preterm and near-term piglets. Anti-microbial defense as measured by neutrophil influx in the colon was only observed in near-term piglets, correlated with histological damage and was reduced by MP-WPC. Moreover, MP-WPC stimulated iALP activity in the colonic epithelium and increased differentiation into enteroendocrine cells compared to EH-WPC. Conclusions. Compared to extensively heated WPC, a formula based on mildly pasteurized WPC limits gut inflammation and stimulates gut maturation in preterm and near-term piglets and might therefore also be beneficial for preterm and (near) term infants.

## 1. Introduction

The human digestive tract is structurally mature at birth, but development of its digestive functions and mucosal barrier functions continues for the next 1–2 years [1,2]. For example, levels of lipases and bile salts, which affect lipid absorption, are lower in infants compared to adults [3]. Further, the mucosal barrier in newborns is characterized by high permeability, lower levels of immunoglobulin A and Paneth cell-derived antimicrobial peptides, and immature responses of mucosal immune cells [2]. Colonization with commensal bacteria and nutrition steer maturation of gut functions [4,5,6]. Conditions like early life undernutrition, intestinal infections, and antibiotics challenge proper gut maturation. Being born preterm provides an extra burden, as preterm infants have a very immature intestine that is less able to deal with challenges, making them more susceptible to clinical complications such as feeding intolerance, intestinal inflammation and necrotizing enterocolitis (NEC) [7,8].

Breastfeeding during the first year of life is widely recognized as the best nutrition for infants, providing many health benefits including early life gut health benefits [9,10,11,12]. Human milk is very complex, providing an optimal nutrition for infants and components with biological activities that drive the maturation of the gut functions. In cases where breastfeeding is not possible, high-quality formulas that meet the complex nutrient requirements of the infant and which stimulate gut maturation and overall health must be available as an alternative. Processing steps such as heat treatments in the production of infant milk formula (IMF) can negatively impact protein nativity and thereby protein bioactivity, and lead to formation of advanced glycation end products [13,14,15,16]. The quality of IMFs might be improved by adapting processing steps and by limiting the heat load. By doing this the bioactive functions of milk proteins, i.e., immune modulation, antimicrobial activity, stimulation of digestive functions and gut barrier functions, are likely to be maintained or less impacted. Previously, it has been shown that heat load and possibly also processing steps in IMF production can be limited by using cold membrane filtration, which results in a whey protein concentrate (WPC) that contains native whey proteins and also (native) beta-casein [17].

Piglets are widely used as a preclinical model in nutritional research, due to the high similarities of the porcine gastro-intestinal tract in digestive physiology, intestinal morphology, metabolism and innate defense to the human gastro-intestinal tract [18,19]. The preterm piglet model is characterized by an extremely immature intestine which is highly sensitive to dietary interventions [16,20]. This high sensitivity toward diet is induced in this model by a combination of preterm birth, cesarean delivery and deprivation of enteral sow’s colostrum, which contains many bioactive peptides that stimulate gut maturation and functioning and protects against gastro-enterocolitis and intestinal infections [21,22]. In addition, by replacing most of the dietary lactose with glucose polymers like maltodextrin which are poorly digested by preterm piglets, a pro-inflammatory state leading to a higher incidence of severe gastro-enterocolitis is induced in this model [23]. Nutritional intervention studies using this established model show extreme prematurity in digestive functions combined with mild to severe gastro-enterocolitis depending on the type of diet [23,24,25].

In this exploratory study, we aimed to investigate the impact of mildly pasteurized WPC (MP-WPC, i.e., heated at 73 °C, for 30 s) obtained by cold filtration versus extensively heated (EH-WPC, i.e., heated at 80 °C for 6 min on top of the mild heat treatment/pasteurization) on intestinal morphology, inflammation and maturation in preterm born piglets. Additionally, a group of near-term piglets was included to investigate whether sensitivity to the dietary intervention depends on gestational age. In five-day nutritional intervention studies the MP-WPC based formula was compared to a EH-WPC based formula. Specifically, we hypothesized that the MP-WPC formula protected against intestinal lesions, limited intestinal inflammation and enhanced gut maturation.

## 2. Materials and Methods

### 2.1. Piglet Study

Preterm and near-term piglets (Danish Landrace x Large White x Duroc) were delivered from sows by caesarean section at 90% gestation (106 days of gestation, n = 34, 2 litters) and 96% gestation (113 days of gestation, n = 18, 1 litter). Piglets were transferred to the intensive care unit and housed individually in heated incubators with air and oxygen supply. Surgical preparation with an orogastric feeding tube and an arterial catheter for parenteral nutrition (PN) and passive immunization took place as previously described [21,22]. The Danish National Committee on Animal Experimentation approved all procedures, which is in accordance with the EU Directive 2010/63/EU Article 23.2 and the Danish executive order no 2014-15-0201-00418. The experimental set-up is described in Appendix A. The WPC used in this study was prepared from raw cow’s milk by cold filtration as described previously [17]. Piglets from each litter were block randomized according to birth weight into two groups of enteral diets: 1) MP-WPC group received formula based on mildly pasteurized WPC (i.e., heated at 73 °C for 30 s) to limit heat load and thereby maintaining protein nativity yet ensuring microbial safety [26,27]; and 2) EH-WPC group received formula with WPC that was, in addition to the mild pasteurization, heat treated at 80 °C for 6 min to maximize the heat load and thereby inducing extensive protein denaturation [26]. Each formula consisted of 80 g/L whey protein concentrate, 50 g/L Pepdite (infant milk formula containing non-milk derived low molecular weight peptides, essential amino acids, carbohydrates, fats, vitamins, minerals and trace elements), 50 g/L Liquigen (medium-chain fatty acids) and 30 g/L Calogen (long-chain fatty acids) (all obtained from Nutricia advanced medical nutrition, Lillerød, Denmark). Macronutrient composition of the formulas was as follows: 3629 kJ/L energy, 59 g/L protein, 52 g/L fat, 39 g/L carbohydrate (of which 21 g/L maltodextrin, 2.7 g/L maltotriose and 1.8 g/L maltose) and 16 g/L lactose. During the study period, piglets in each group received the enteral nutrition in increasing dose, starting with 6 mL/kg/3 h at day 1, increasing to 8 mL/kg/3 h at day 2 and 3 and 10 mL/kg/3 h at day 4 and 5. In addition to the enteral nutrition, all piglets received 4 mL/kg*h PN from day 1 to day 5, which is in the range of previous used volumes of EN in this model [15,16]. PN solution was based on a commercially available product (Kabiven, Fresenius Kabi, Uppsala, Sweden) and adjusted in nutrient composition to meet the requirement of piglets. For passive immunization, piglets received three doses of maternal serum (iv). On day 5, intestinal tissue was collected. For this, the piglets were first anaesthetized with an intramuscular injection of Zoletil mix (Tiletamine and Zolazepam, 0.1 mL/kg, Virbac, Kolding, Denmark) and subsequently 5 mL of 20% pentobarbital (Euthanimal, Scanvet, Denmark) was injected intracardiac to euthanize the piglet, compliant with the ethical guidelines described above.

### 2.2. Clinical Evaluation and Sample Collection

Piglets were continuously monitored by caretakers and veterinarians during the entire study. Piglets were euthanized on day 5 for tissue collection, or when clinical symptoms of gastro-enterocolitis (feeding intolerance, vomiting, abdominal distention, hemorrhagic diarrhea and/or respiratory distress) appeared during the study and the humane endpoint was reached. Tissues of three regions of small intestine (proximal, middle and distal) and colon were macroscopically evaluated to assess severity of gastro-enterocolitis blinded for dietary intervention (see Appendix A and as described previously in [22]).

### 2.3. Microscopic Evaluation

Paraformaldehyde-fixed tissues from distal small intestine and colon were processed for histology (hematoxylin-eosin staining) to determine microscopic tissue damage. Tissue was evaluated for the presence of edema, the integrity of the epithelium, location of erythrocytes, immune infiltration and villous atrophy (Appendix A). Parameters were scored blinded and grades for each parameter were added up to result in a total histology score ranging from 0 to 15 for distal small intestine, and 0 to 12 for colon. excluding villous atrophy.

### 2.4. Immunohistochemistry

To localize proliferating epithelial cells (Ki67), enterochromaffin cells (5HT) and immune cell influx (CD3, S100A9, MPO, MMP9), immunohistochemistry was performed as described previously [28]. In short, tissue slides were deparaffinized with xylene and gradually rehydrated in ethanol. After antigen retrieval in sodium citrate buffer (20 min at 100 °C) and 1 h blocking (1% BSA 0.1% Triton X-100 in PBS), slides were incubated overnight with primary antibody in blocking buffer. Staining was visualized by 30 min incubation with Powervision secondary antibody (Immunologic, VWR, Amsterdam, The Netherlands) and subsequent 10 min chromagen substrate diaminobenzidine (Sigma-Aldrich, Zwijndrecht, the Netherlands). The following antibodies were used: rabbit polyclonal anti-CD3 (1:2000, DAKO, Agilent Technologies A0452, Amstelveen, the Netherlands), rabbit polyclonal anti-Ki67 (1:8000, Abcam, ab15580, Cambridge, UK), rabbit polyclonal anti-MMP9 (1:1000, Abcam, ab76003, Cambridge, UK), rabbit polyclonal anti-MPO (1:100, Abcam, ab9353, Cambridge, UK), mouse monoclonal anti-S100A9 (1:500, ThermoFisher Scientific MAC387, Bleiswijk, the Netherlands) and mouse monoclonal anti-5HT-H209 (1:100, ThermoFisher Scientific MA5-12111, Bleiswijk, the Netherlands). Mucus-containing cells were visualized by periodic acid-Schiff’s (PAS) staining. Intestinal alkaline phosphatase (iALP) brush border activity on tissue slides was determined with NBT/BCIP conversion as described previously [29,30]. In short, after deparaffinization and rehydration, slides were incubated with NBT/BCIP (Sigma-Aldrich, Zwijndrecht, the Netherlands) in NTM buffer (0.1 M Tris, 50 mM MgCl_2_, 0.1 M NaCl, pH 9.5) for 30 min. After counterstaining with nuclear fast red (Sigma-Aldrich, Zwijndrecht, the Netherlands), slides were dehydrated and mounted. Images were processed using Olympus BX51 microscope and quantified with ImageJ (version 1.52a, National Institutes of Health, US). Relative staining intensity was quantified blinded in representative images of selected area, 3 images per piglet at 20 × magnification.

### 2.5. Gene Expression Analysis

Messenger RNA was isolated from the colon to determine gene expression levels of the gut maturation markers and intestinal inflammation markers by quantitative-PCR analysis. In brief, RNA was isolated with TRI-reagent (Sigma-Aldrich, Zwijndrecht, The Netherlands) and purified with the Bioline ISOLATE II RNA Mini kit (BIO-25073, Bioline, London, UK) according to manufacturer’s protocol. Using Revertaid reverse transcriptase (Fermentas, Vilnius, Lithuania), 1.0 μg of RNA was transcribed. Quantitative RT-PCR was performed on a BioRad iCycler using Sensifast SYBR No-ROX Kit (GC-biotech Bio-98020, Waddinxveen, The Netherlands) according to manufacturer’s instructions. From a panel of six genes, most stable reference genes were identified by GeNorm algorithm [31]. Relative expression levels were calculated with N0 values obtained by LinRegPCR, and normalized to reference genes GAPDH and RPL4. Negative controls without reverse transcriptase was included to verify the absence of contaminating genomic DNA. Primers used were designed for pig and specificity was validated based on melting curves and product size. Sequences are provided in Appendix A.

### 2.6. Enzyme Activity Assay

Snap-frozen tissue from the small intestine or colon was homogenized and activity of intestinal alkaline phosphatase (iALP) (EC 3.1.3.1) was determined by spectrophotometry with a diethanolamine assay measuring pNPP hydrolysis according to manufacturer’s instruction (Phosphatase substrate, ThermoFisher Scientific, Bleiswijk, the Netherlands) and as previously described [32]. Specificity of the assay for intestinal-specific form of ALP was confirmed by pre-incubating samples with L-phenylalanine [33]. Enzyme activity values were corrected for total amount of protein in the samples, as determined by BCA reaction, and are expressed as µg pNPP/mg protein·min^−1^.

### 2.7. Protein Analysis

Levels of IL-8 and TNF-α in homogenized intestinal tissue were determined with porcine IL-8/CXCL8 DuoSet ELISA (DY535, R&D Systems, Abingdon, UK) and porcine TNF-alpha DuoSet ELISA (DY690B, R&D Systems, Abingdon, UK) according to manufacturer’s instructions, in 50 µL undiluted sample. Cytokine levels were corrected for protein concentration in the samples, as determined by BCA.

### 2.8. Statistical Analysis

This was an exploratory study with limited number of animals and therefore limited power, especially for the near-term piglets. Six animals of one preterm litter experienced severe clinical symptoms of gastro-enterocolitis and had to be euthanized before day 5 (according to humane end-point guidelines). Limited exposure to the diet (ranging from 2–4 days) in these drop-out piglets might influence specific dietary outcomes, but all observations remained similar when excluding these animals. Furthermore, comparing preterm litter 1 with litter 2 did not reveal any litter effect. Therefore, the obtained data of all preterm piglets were included in all analysis. One near-term piglet had to be euthanized according with the humane end-point due to sepsis related to catheter issues and was excluded from analysis. Statistical analysis was performed using GraphPad Prism version 8 software (La Jolla, CA, USA). Data are expressed as median with interquartile range (IQR) since most data was not normally distributed (determined by D’Agostino test). Clinical symptoms (i.e., bodyweights, organ weights, first time standing) were compared between both diets and gestational ages by a Kruskall-Wallis test, with Dunn’s post-test for specific multiple comparisons. For most other parameters evaluated in this study, the effect of diet was tested only within the different gestational age groups, comparing EH-WPC with MP-WPC by a Mann-Whitney test. The correlation between macroscopic and histology scores was determined by Spearman’s correlation test and the difference in % of piglets with ALP positive brush border was tested by Fisher’s Exact test. P values below 0.05 were considered statistically significant.

## 3. Results

### 3.1. Clinical Symptoms

Preterm piglets had significantly lower birth weights compared to near-term piglets (944 gr ± 305 vs. 1183 g ± 463, *p* = 0.004) and required longer time till standing up on their feet after birth (26 h ± 8 vs. 6 h ± 17, *p* < 0.001) (Appendix A). Irrespective of type of diet, preterm piglets showed a significantly lower weight gain over the 5-day intervention period than near-term piglets (15% vs. 10.5%, *p* = 0.013). These findings show that preterm piglets respond differently after birth compared to near-term piglets.

An effect of the dietary intervention on postnatal growth was measured by determining the weight of the gastro-intestinal tract, liver and spleen relative to bodyweight. While liver, small intestine and colon weight were not affected, the weight of the empty stomach was significantly lower for piglets fed MP-WPC compared to EH-WPC. In addition, relative spleen weight was lower in piglets fed MP-WPC compared to EH-WPC (Appendix A).

In one of the two preterm litters, several cases of feeding intolerance, abdominal distention, hemorrhagic diarrhea and/or respiratory distress were observed during the study. Six animals experienced severe clinical symptoms of gastro-enterocolitis and had to be euthanized before day 5. Of these 6 animals, 3 received EH-WPC and 3 received MP-WPC, indicating the gastroenterocolitis was not related to the diet, but most likely related to the extreme prematurity or potential hypoxic state during caesarean section of this litter. Clinical symptoms of gastroenterocolitis were less pronounced in the near-term piglets compared to the preterm piglets during the intervention, with no signs of feeding intolerance observed during the 5-day intervention. In summary, preterm piglets responded differently to a dietary intervention compared to near-term piglets; however, no distinct difference in growth or diet tolerance between EH-WPC and MP-WPC was observed.

### 3.2. Intestinal Morphology

At day 5, macroscopic signs of gastroenterocolitis in the intestinal tract (score 1–6) were mostly absent in the proximal, middle and distal small intestine in preterm as well as near-term piglets (Figure 1a–c), whereas a high degree of lesions was observed in the colon, especially in preterm-born piglets (Figure 1d). Interestingly, colon lesion scores were limited in MP-WPC fed near-term piglets (median score 1.0 ± 0.75) compared to EH-WPC (median score 3.0 ± 2.0) (Figure 1d). A similar trend was seen in preterm piglets, with a median macroscopic score of 2.0 ± 3.5 for MP-WPC compared to 3.0 ± 2.0 for EH-WPC (Figure 1d).

The effect of MP-WPC and EH-WPC with respect to microscopic lesions was studied in detail in the distal small intestine and colon. A histology damage scoring system was developed based on lesions observed in infants and piglets with NEC [34,35], and previous experience with systematic scoring of colitis in mice [36] (see Appendix A). Lesions included various degrees of edema, epithelial disruption, presence of erythrocytes, immune cell infiltration and, for the small intestine, different levels of villus atrophy (Figure 1e). Microscopic evaluation of histological damage revealed a similar pattern observed after macroscopic evaluation, displaying no dietary effect in the distal small intestine (Figure 1f). However, lower total histological damage scores in the colon of piglets fed MP-WPC in comparison to piglets fed EH-WPC were noted (Figure 1g). Additionally, there was a significant positive correlation between the macroscopic and histological damage scores for both preterm (Spearman r = 0.55, *p* = 0.0016) and near-term piglets (Spearman r = 0.83, *p* < 0.0001). In the distal small intestine, mainly villus atrophy accounted for the histological damage in addition to minor immune cell infiltration. In the colon all four parameters contributed to the total histology score in both diet groups as well as in preterm and near-term piglets.

Results of both macroscopic and microscopic evaluation of intestinal morphology indicated that a diet based on MP-WPC limited colonic damage compared to a diet based on EH-WPC in preterm and near-term piglets.

### 3.3. Intestinal Inflammation

Damage to the intestinal barrier and potential bacterial translocation is associated with an acute inflammatory response, therefore the inflammatory status of the colon was next evaluated by RT-qPCR. Irrespective of diet, cytokine RNA expression levels in the colon of preterm piglets were higher than in near-term piglets. Except for colonic *TNF-α* levels in preterm piglets, expression levels of *IL-1β* (Figure 2a), *IL-8* (Figure 2b) and *TNF-α* (Figure 2c) were lower in piglets fed the MP-WPC diet compared to EH-WPC, indicating lower grade of colonic inflammation. Protein levels of IL-8 and TNF-α confirmed the gene expression patterns (Appendix A). Other cytokines associated with inflammation in the immature intestine were not affected by type of WPC (*IL-18*, Appendix A) or undetectable (*IL-22*).

To study the inflammatory response in the colon in more detail, influx of immune cells was evaluated by CD3 immunohistochemistry for T cells (Figure 2d). T cells infiltration in colonic submucosa was observed in both preterm and near-term piglets. Interestingly, less T cell infiltration was noted in piglets receiving MP-WPC in both preterm and near-term piglets (Figure 2e). Accordingly, *CD14* expression was also significantly lower in MP-WPC fed preterm and near-term piglets (Figure 2f) and a similar trend was observed for *TLR4* expression (Figure 2g). Both receptors are present on antigen-presenting cells but can also be found on colonic intestinal epithelial cells. Activation of these receptors by lipopolysaccharide (LPS) results in activation of downstream inflammatory signaling pathways. Our data, based on the cytokine profile and influx of T cells, suggest that colonic inflammation in the immature intestine is limited after exposure to MP-WPC compared to EH-WPC.

In the distal small intestine, i.e., ileum, there were minimal signs of inflammation, with lower levels of *IL-1β* in MP-WPC compared to EH-WPC fed preterm piglets (Appendix A), while *IL-8* and *TNF-α* levels remained unaffected (Appendix A). T-cells were present in the distal small intestinal crypt region (Appendix A), however no significant differences in T cell infiltration between type of diet or gestational ages were observed (Appendix A).

### 3.4. Innate Defense

In addition to T-cells, neutrophils play an important role in the inflammatory response. As part of the innate defense system, neutrophils can produce antimicrobial peptides in response to bacterial translocation, including myeloperoxidase (MPO), matrix metallopeptidase-9 (MMP9) and calprotectin (dimer of S100A9-S100A8). Presence of neutrophils in the colonic tissue was identified by immunohistochemistry for these antimicrobial peptides. Irrespective of diet, staining for all three markers was nearly absent in the preterm piglet colon, while positive cells were identified in the near-term piglet colon (Figure 3a). A distinct dietary effect on neutrophil influx was observed in the near-term piglets. The presence of MPO^+^ cells in near-term piglets was significant lower upon a diet with MP-WPC compared to a diet with EH-WPC (Figure 3b). These findings were confirmed by less staining of MMP9 (Figure 3c) and S100A9 (Figure 3d). Interestingly, there was a significant positive correlation between staining intensity and the histological damage scores for both preterm (Spearman r = 0.69, *p* < 0.0001) and near-term piglets (Spearman r = 0.53, *p* = 0.032). In the distal small intestine, MPO and MMP9 were not detected, while the low level of S100A9 protein was lower in near-term piglets fed MP-WPC compared to EH-WPC (Appendix A).

Next, the innate defense provided by the intestinal epithelial cells and the influence of diet on this function was evaluated. Intestinal alkaline phosphatase (iALP) is expressed by colonic enterocytes and facilitates protection from microbial stimuli by dephosphorylating LPS. Staining with the substrate NBT/BCIP to localize iALP activity showed specific activity of the enzyme at the brush border of the colon epithelium (Figure 3e). The percentage of preterm piglets positive for iALP activity in colon tissue was 76% for MP-WPC fed piglets and only 20% of EH-WPC fed piglets (Figure 3f). A similar trend was observed for near-term piglets fed MP-WPC, with 56% positive for EH-WPC and 100% positive for MP-WPC. The increased activity of iALP in the colon was further confirmed by enzyme activity assay (Figure 3g). iALP activity levels measured did not correlate to total histology damage score for both preterm and near-term piglets. Together, these results show that MP-WPC can maintain or increase alkaline phosphatase activity in the colon, likely contributing to an improved innate defense. In the distal small intestine, all piglets showed iALP positive brush border (Appendix A) with no differences between diets on enzyme activity level (Appendix A).

### 3.5. Epithelial Functioning: Cell-Type Specific Differentiation and Proliferation

In addition to intestinal inflammation, the impact of EH-WPC and MP-WPC on epithelial cell proliferation and differentiation was evaluated. Proliferating cells in the colon crypt were identified by immunohistochemical staining for Ki67 (Figure 4a) and quantification of the number of Ki67^+^ cells per crypt showed a lower proliferation with a diet based on MP-WPC compared to EH-WPC in both preterm and near-term piglets (Figure 4b). Measurement of colon crypt depth showed a smaller crypt depth in both preterm and near-term piglets receiving MP-WPC compared to EH-WPC (Figure 4c). Expression level of *OLFM4*, a marker for intestinal stem cells and transit amplifying cells, was lower in near-term piglets fed MP-WPC compared to EH-WPC (Figure 4d), but levels were not different in preterm piglets.

Next, goblet cells in the colonic epithelium were evaluated. mRNA levels of *MUC2* showed no significant changes when exposed to different diets (Figure 4e). Yet, staining of mucus with PAS revealed overall more intact goblet cells in MP-WPC with more PAS-positive mucus-containing granules (Figure 4f). *MUC1*, a cell membrane-bound mucin expressed on enterocytes to serve as an additional layer of defense, was unaffected by diet (Figure 4g). Villin 1 (*VIL1)* expression levels, present in the brush border of all intestinal epithelial cells, tended to be lower in colon exposed to MP-WPC (Figure 4h). Intestinal fatty acid binding protein2 (*FABP2*) and carbonic anhydrase2 (*CA2*), markers for differentiated colon enterocytes, were not detected in the immature colon of both preterm and near-term piglets.

Finally, enterochromaffin cells in the colon were detected by staining for serotonin (5HT) (Figure 4i). Interestingly, quantification of the number of 5HT^+^ cells per crypt showed an increase due to gestational age/maturation status (i.e., near-term vs. preterm) and an additional increase when exposed to MP-WPC compared to EH-WPC (Figure 4j). Together, these findings imply that a diet based on MP-WPC limits epithelial hyper proliferation compared to EH-WPC, and thereby improves/allows epithelial cell differentiation and likely epithelial maturation.

Epithelial cell functioning was also evaluated in the distal small intestine. Staining of goblet cells by PAS showed no differences between EH-WPC and MP-WPC (Appendix A), and *MUC2* expression profiles also did not show differences between diet groups (Appendix A). *FABP2* showed higher expression in the near-term distal intestine compared to the preterm intestine but was unaffected by diet (Appendix A). Markers for stem cells and transit amplifying cells (*OLFM4*), Paneth cells (*LYZ*) and enteroendocrine cells (*CHGA*) displayed similar expression levels for EH-WPC and MP-WPC (Appendix A), with *LYZ* expression levels significantly higher in near-term piglets compared to preterm piglets.

## 4. Discussion

The preterm piglet model is widely used for nutritional intervention studies, as it is a model that is hypersensitive to enteral nutrition [16,19,20]. In the current study, the effect of MP-WPC and EH-WPC on gut histology, inflammation and innate defense in preterm and near-term piglets was evaluated. A 5-day enteral diet based on MP-WPC caused fewer intestinal lesions, less acute inflammation and improved innate defense in preterm and near-term piglets, compared to EH-WPC. Additionally, the degree of prematurity, i.e., preterm versus near-term, caused differences in the intestinal immune response and the intestinal epithelial maturation/differentiation response.

Detailed macroscopic and microscopic analysis of the gastro-intestinal tract revealed that in both the preterm and the near-term piglets the colon was the most affected region and that the macroscopic and microscopic degree of intestinal lesions were highly correlated. Total histology score gives detailed insight into the type of lesions and the degree of damage, yet it does not take into account the focal occurrence of the gastroenterocolitis. Combining these two types of analysis gave a more complete picture of the overall intestinal injury as demonstrated in our study.

As the consequences of inappropriate functioning of the intestinal barrier are likely first observed in the colon due to the high bacterial load, and as colon was the most affected by diet in our study, we focused our analyses of the dietary intervention primarily on colonic tissue. Two major findings related to the inflammatory response in the colon were: (1) MP-WPC diet limited the inflammatory response compared to EH-WPC; and (2) preterm piglets showed overall a higher cytokine response compared to near-term piglets. More specifically, gene expression of *IL-1β*, mainly expressed by innate immune and epithelial cells, was lower in preterm piglets fed MP-WPC than EH-WPC. Interestingly, dietary intervention with MP-WPC limited *TNF-α* expression in near-term piglets, while levels remained high in preterm piglets. The main producers of TNF-α are T cells, and an influx of T cells in the colonic submucosa of near-term piglets was indeed identified. TNF-α-producing T cells in the immature human intestine have been shown to mediate inflammation upon preterm birth [37]. The differences in TNF-α levels between preterm and near-term piglets likely reflect a difference in immaturity of the immune system between preterm and near-term piglets.

Of all evaluated cytokines, the IL-8 response was the most robust in both preterm and near-term piglets and, therefore, likely a key cytokine in the inflammatory response observed. The production of IL-8 by enterocytes is developmentally regulated [38,39], and in preterm infants it is used as biomarker of onset and severity of NEC disease [40,41,42]. IL-8 is a chemokine that attracts neutrophils to site of inflammation. Remarkably, a neutrophil influx was only observed in the near-term piglets, while being absent in the preterm piglets, potentially reflecting immaturity of the immune system in the preterm piglet. Despite causing local damage, neutrophils have an important function in the innate defense by producing antimicrobial molecules [43]. A lack of a proper neutrophil response potentially results in less protection and a higher susceptibility to intestinal damage. In preterm infants, research has identified a limited neutrophil precursor pool, reduced neutrophil phagocytosis and impaired neutrophil extracellular trap formation [44,45,46,47]. However, impaired neutrophil infiltration in intestinal tissue due to immaturity has not been described in preterm infants. On the contrary, high levels of neutrophil MP-WP marker calprotectin have been found in the stool and tissue of preterm infants with NEC [48,49]. Results presented here imply impaired migration of neutrophils to the colon of preterm piglets or an inability of the mucosal neutrophils to produce anti-microbial molecules. These data suggest that preterm piglets are even more immature than preterm infants with respect to the mucosal neutrophil response.

Our data suggest that MP-WP limits intestinal inflammation and associated damage in preterm piglets via iALP activity, which was increased in the colon exposed to MP-WPC compared to EH-WPC. The increased iALP activity can mediate a decrease in CD14/TLR4 stimulation by LPS, resulting in less synthesis and secretion of IL-8 by the immature epithelium and thereby prevention of (excessive) inflammatory responses and promotion of mucosal tolerance [50]. In mice, body-wide deletion of iALP results in increased inflammation, permeability and bacterial translocation in the newborn intestine [51]. The expression and activity of iALP can be influenced by nutrition and the intestinal microbiota, but is also dependent on maturation status of the intestine [52,53]. Intestinal ALP activity levels measured in this study did not correlate to total histology damage scores, suggesting the increase in enzyme activity is a result of dietary intervention rather than a secondary effect of the epithelial damage. These data warrant further research to obtain more mechanistic insights into the dietary effects on iALP activity.

It appeared that specific epithelial cell functions were affected by the inflammation secondary to dietary intervention. Although no changes were measured on mRNA level for the goblet cell marker *MUC2*, we did observe differences in mucin containing goblet cells (i.e., PAS-positive) suggesting mucin hypersecretion. Specifically, in intact colon crypts, loss of mucin containing goblet cells was identified in preterm as well as near-term piglets fed EH-WPC based formula. In preterm infants, the initial response to inflammation is upregulation in the number of mucin containing goblet cells, followed by mucin hyper-secretion and at the final stage depletion of mucin-positive goblet cells [54,55,56]. Furthermore, higher mucus secretion in the EH-WPC treated piglets might be, at least partly, mediated via the increased TNF-α levels measured in the colonic tissue, which has been shown to increase mucin secretion in the immature intestine of preterm infants with NEC [57].

In response to inflammatory signals and intestinal damage, epithelial cells can start to hyper proliferate, in order to regenerate the epithelium [37]. In our study, epithelial proliferation was reduced in colonic epithelium exposed to MP-WPC compared to EH-WPC, with implications for epithelial-specific cell differentiation. Immunohistochemistry identified an increase in serotonin^+^ cells in the colon of piglets fed MP-WPC diet compared to EH-WPC. These enterochromaffin cells are a subset of enteroendocrine cells which, besides regulating gastro-intestinal motility, also play a role in sensing microbial metabolites. Serotonin signaling has been described as maturing with increasing postnatal age [58], and effects observed here might therefore have important implications for gut motility and more generally the development and functioning of the gut-brain axis.

## 5. Conclusions

In summary, a formula based on mildly pasteurized WPC has beneficial effects on colonic inflammation and maturation compared to extensively heated WPC in preterm and near-term piglets and might therefore also enhance intestinal maturation and limit gut inflammation in preterm and (near-) term infants. Similar responses to the dietary intervention were found in preterm and near-term piglets considering lesion scores, intestinal inflammation and iALP activity, but the neutrophil response and enterochromaffin cell differentiation response appeared to be specific for the near-term piglets. Yet in the current study only one near-term litter was included, which limits the power and interpretation of the outcomes. Considering this, future studies on the influence of diet on gut maturation should perhaps also focus on (near-) term piglets. Besides, more mechanistic studies in vitro are justified to identify specific pathways involved in these effects.

## Figures and Tables

**Figure 1 nutrients-12-01125-f001:**
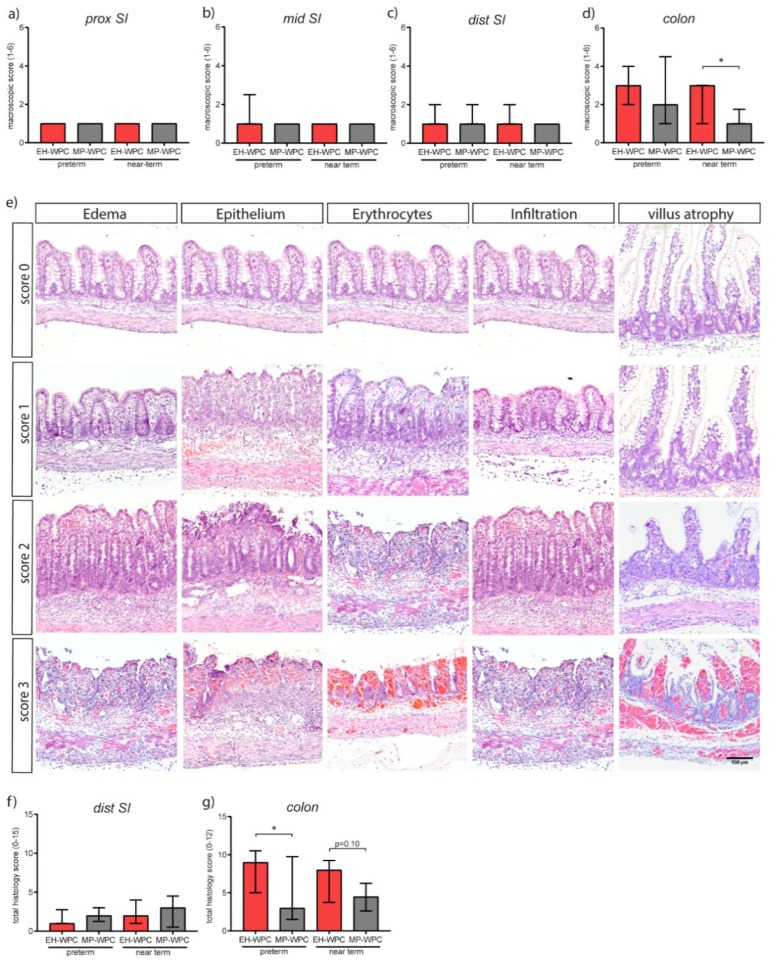
Mildly pasteurized whey protein concentrate (MP-WPC) limits colonic lesions. After 5 days of enteral nutrition, the gastro-intestinal tract of preterm and near-term piglets was macroscopically scored for signs of gastroenterocolitis in (**a**) proximal small intestine, (**b**) middle small intestine, (**c**) distal small intestine and (**d**) colon. (**e**) Representative images of various degrees of lesions observed, including edema, epithelial damage, presence of erythrocytes, infiltration of inflammatory cells and villus atrophy. Tissue was further evaluated microscopically with a total histology score for (**f**) distal small intestine and (**g**) colon. Values are median ± IQR, * *p* ≤ 0.05. Scalebar equals 100 µm.

**Figure 2 nutrients-12-01125-f002:**
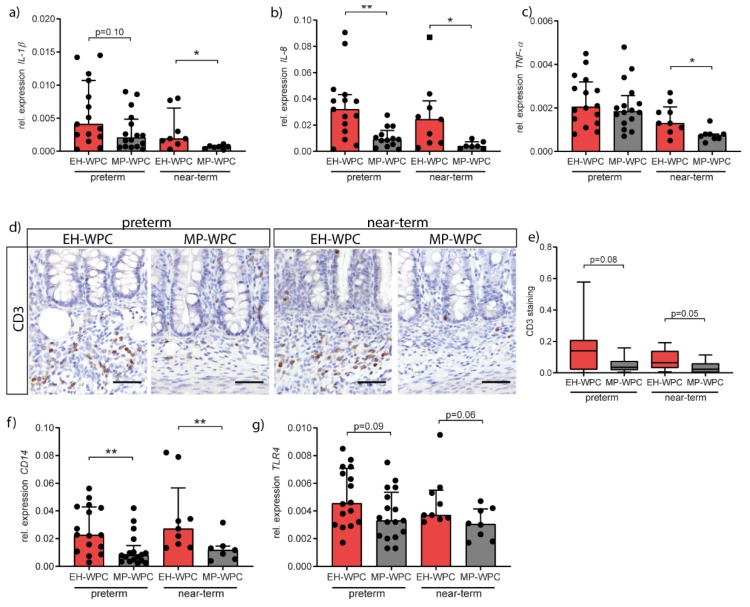
Colonic inflammation is limited in immature intestine exposed to MP-WPC compared to extensively heated (EH-)WPC. Relative expression levels of inflammatory cytokines (**a**) *IL-1β*, (**b**) *IL-8* and (**c**) *TNF-α* in the colon as determined by RT-qPCR. (**d**) T cells were detected in colon tissue by immunohistochemistry staining for CD3, and (**e**) staining intensity in the crypt region was quantified by ImageJ analysis. Relative expression levels of (**f**) *CD14* and (**g**) *TLR4* in the colon as determined by RT-qPCR. Values are median ± IQR, * *p* ≤ 0.05 ** *p* ≤ 0.01. Scalebar equals 50 µm.

**Figure 3 nutrients-12-01125-f003:**
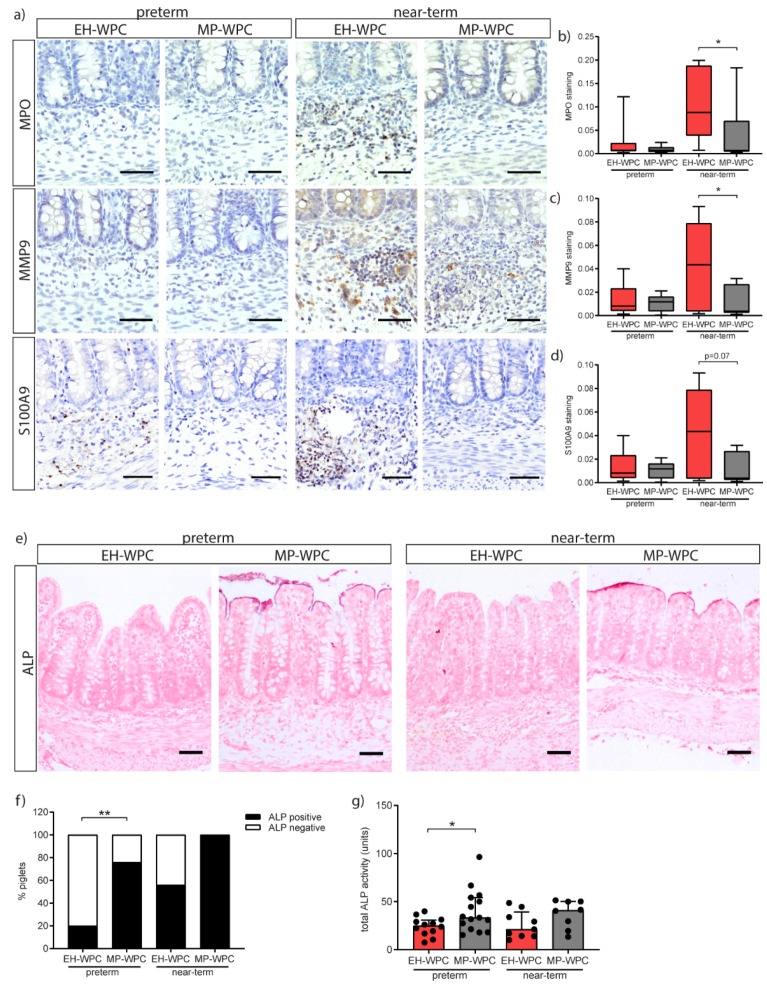
Innate defense in the colon is influenced by MP-WPC and gestational age. (**a**) detection of neutrophils by immunohistochemistry in colon tissue, with relative staining intensity of (**b**) MPO, (**c**) MMP9 and (**d**) S100A9 quantified by ImageJ analysis. (**e**) iALP brush border activity on colon tissue slides was determined with NBT/BCIP conversion and (**f**) number of piglets with iALP positive brush border was quantified. (**g**) Total iALP enzyme activity (µg pNPP·mg protein·min^−1^) determined on protein level in colon tissue homogenates and content. Values are median ± IQR, * *p* ≤ 0.05 ** *p* ≤ 0.01. Scalebar equals 50 µm.

**Figure 4 nutrients-12-01125-f004:**
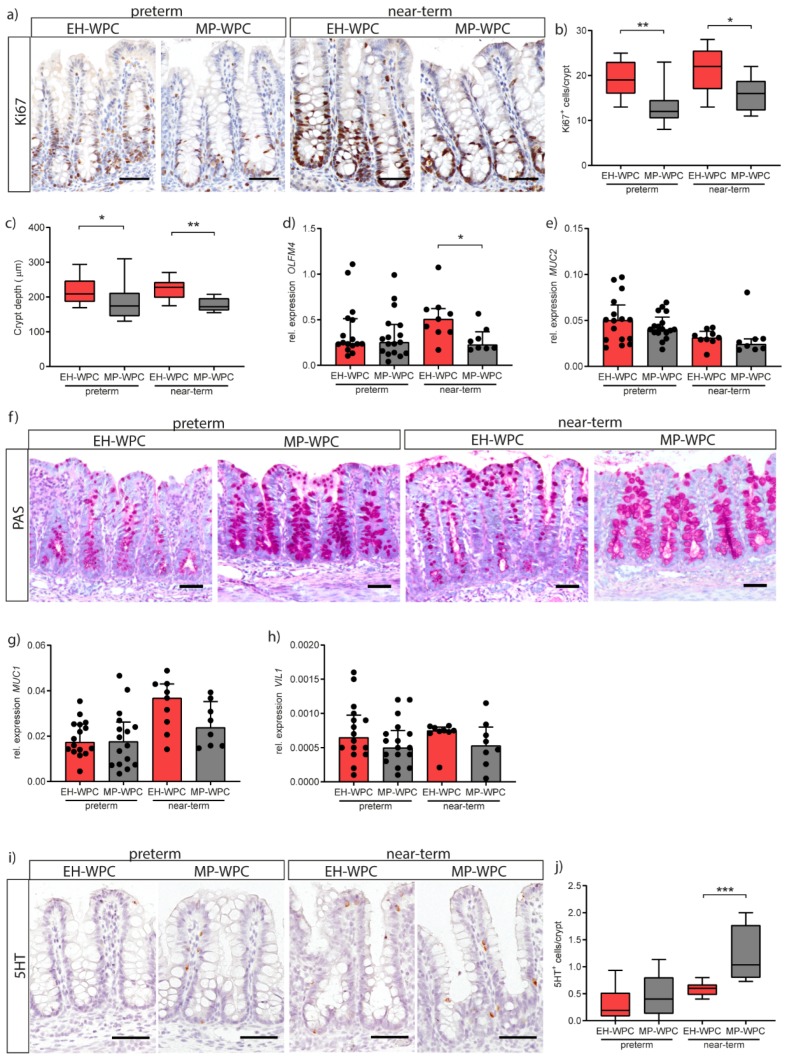
MP-WPC affects colonic crypt proliferation & differentiation. (**a**) Proliferating cells in the colon crypt were identified by immunohistochemistry staining for Ki67 with (**b**) quantification of the number of Ki67^+^ cells per crypt. (**c**) Colon crypt depth evaluated by ImageJ analysis. Relative expression levels of stem cell marker (**d**) *OLFM4* and goblet cell marker (**e**) *MUC2* as determined by RT-qPCR and (**f**) detection of mucus-containing cells in the colon by Periodic-Acid Schiff’s (PAS) staining. Relative expression levels of (**g**) *MUC1* and (**h**) *VIL1* in the colon as determined by RT-qPCR. (**i**) enterochromaffin cells in de colon were detected by immunohistochemistry staining for serotonin (5HT) and (**j**) number of 5HT^+^ cells per crypt were quantified. Values are median ± IQR, * *p* ≤ 0.05 ** *p* ≤ 0.01 *** *p* ≤ 0.001. Scalebar equals 50 µm.

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
