# Peer review of "Beneficial Effect of Mildly Pasteurized Whey Protein on Intestinal Integrity and Innate Defense in Preterm and Near-Term Piglets"

_nutrients, 2020, doi:10.3390/nu12041125_

Round 1

Reviewer 1 Report

This study in newborn piglets compared the effects on the gut of a mildly heat treated whey concentrate with the effects of an extensively heat treated (standard) whey concentrate. The mildly treated concentrate had fewer adverse effects on the newborn gut. This finding has relevance with regard to formulas that are sometimes needed for premature infants when mother's milk, for one reason or another, is not available.

The study was well designed and expertly conducted by investigators working with members of a group (led by Sangild) that has performed similar studies for some time. The newborn piglet, premature as well as close to term, is a proven experimental animal for this type of question.

Some minor comments:

Line 27: change "histology" to "histologic"

Line 36: change "beneficial" to "fewer adverse"

Line 41: List of abbreviations is incomplete

Line 65: change "head" to "heat"

Line 66: change "stimulations" to "stimulation"

Line 74: change "extreme" to "extremely"

Line 107: explain the terms Pepdite, Liquigen and Calogen

Line 129: change "on" to "for"

Line 215: change "different" to "differently"

Line 239: change "indicate" to "indicated"

Lines 305 & 306: delete "preterm" x2

Line 342: change "amount" to "number"

Line 382: change "to" to "on"

Line 386: change sentence starting with "was lower" to "was low with MP-WPC         and in preterm piglets".

Lines 405-406: change sentence starting with "On the contrary" to "On the       contrary, high levels of the neutrophil marker calprotectin have been...

Line 443: change "support" to "enhance"

Line 444: change "for" to "in"

Line 446: change "appears" to "appeared"

Reviewer 2 Report

This is a well written paper with clearly presented data. The manuscript has translational value, as it may have implications for a better understanding of the pathogenesis and treatment of necrotizing enterocolitis. However, I do have several concerns.

What is the rationale for using only one litter of preterm piglets, while two litters of near term piglets were used? What if this one particular sow from which the preterm piglets were collected had particular idiosyncratic abnormalities affecting her pregnancy, which could affect her piglets intestinal development? Please discuss the rationale for using two near term litters and only one preterm litter.

The study is underpowered. There are several instances where a trend is evident, but statistical significance is not reached (for example, the comparison of macroscopic features  between EH-WPC and MP-WPC piglet colons in Figure 1D, the lymphocyte study in preterm piglets in Figure 2E, the ALP study in near term piglets in Figure 3G, and the data in Figure 4H). This would probably be less of a problem if a second litter had been included in the preterm group. Please acknowledge this shortcoming and discuss why only one litter was studied in the preterm group.

Why is only gene expression data presented for IL-1B? Altered gene expression does not necessarily translate to altered protein levels. The statement about altered IL-1B production on Page 12, Lines 385-387, is not supported by the data.

Based on Figure 2C, I don't think you can conclude that TNF-a is reduced in MP-WPC piglets compared to EH-WPC piglets.

Figures 4E-4G are very puzzling. Why is the change in PAS staining so striking, while there are no significant differences in either MUC1 or MUC2 expression?

Line 299 on Page 8 says, "...fed MP-WPC compared to MP-WPC..." Is this a typo?

Reviewer 3 Report

The important of enteral feeding in newborns is undisputed and the infant formula is extremely useful and extensive used in the neonatal intensive care unit. However, the thermal process of this nutrition is low explore in the research field and, more interesting, the effect on their components by this physical process is one of the most concerns in the neonatal nutritional industry. In this article, the authors explored this thermal process in the enteral feeding in one animal model very useful more related with preterm infants and their gut development. Extremely useful to increase the knowledge in this field. However, some minor recommendations I would to write to improve the quality of this article to be consider by the authors.

Introduction: in general, this section is well-done wrote. The background is well established and the objectives well described. However, in my opinion, the authors could improve the section if they report their hypothesis or what they expect to found.

Material and methods:

  • Piglet study: the authors could report more details according to, how was the oxygen supply, how was the orogastric tube? The enteral volume intakes used were in normal range?
  • Microscopic evaluation: the score reported was very useful and well-described, however, if was reported previously the authors could give the references (34,35,36) or some parameters of previously validation.
  • Statistical analysis: the survival analysis could be useful to determinate the first paragraph related to died number of the animals. The litter effect could be a limitation, because the litter could be a confounder factor in the analysis. How the authors could justify this aspect? The D’Agostino test could not be the most familiar test to determinate the normality, is this test similar to others? The correlation used should also report. The test for figure 3f had not been reported.

Results:

  • Line 200: the authors did not measurement the weight gain, the measurement the weight change. However, it could be useful to measure as a g/kg/day.
  • Paragraph between 207 to 216, I want to thank to the authors the summary for the feeding intolerance and the interpretation between term and preterm piglets.
  • Intestinal morphology: the score is a number, however the interpretation of this variable could be different, because it is not possible to obtain an animal with 2.33 score. In this situation, maybe the data could be report by percentage and test by Fisher. Would be the results similar?
  • The figure 1e is very useful. However, the report of IQR is no symmetric with median, for this reason, the figures with median and IQR should be with box-plots.

Discussion: In general, the section is well established and described. I want to thank the authors for the effort. The paragraph between lines 367 to 373 is well write and is very useful.

Specific comments referring to line numbers, tables or figures.

  • Line 63: the acronyms “IMF” was not defined before.
  • Line 69: the acronyms “WPC” was not defined before.
  • Line 116: Zoletil is a mix with tiletamine and Zolazepam, Maybe is useful to report it.
  • In general, the manufactures could be complemented with the city and countries.
  • Take to account the decimal separation could be “.” Instead of “,”
  • The acronyms of international unit for grams is “g” no “gr”
